# Two noncompeting human neutralizing antibodies targeting MPXV B6 show protective effects against orthopoxvirus infections

Runchu Zhao[1,8], Lili Wu[1,8], Junqing Sun[1,2,8], Dezhi Liu[1,3,8], Pu Han[1], Yue Gao[1,4], Yi Zhang[1,3], Yanli Xu[5], Xiao Qu[1], Han Wang [6], Yan Chai[1], Zhihai Chen [5], George F. Gao [1,2,7] & Qihui Wang [1,3,7] ✉

The recent outbreak of mpox epidemic, caused by monkeypox virus (MPXV), poses a new threat to global public health. Here, we initially assessed the preexisting antibody level to the MPXV B6 protein in vaccinia vaccinees born before the end of the immunization program and then identified two monoclonal antibodies (MAbs), hMB621 and hMB668, targeting distinct epitopes on B6, from one vaccinee. Binding assays demonstrate that both MAbs exhibit broad binding abilities to B6 and its orthologs in vaccinia (VACV), variola (VARV) and cowpox viruses (CPXV). Neutralizing assays reveal that the two MAbs showed potent neutralization against VACV. Animal experiments using a BALB/c female mouse model indicate that the two MAbs showed effective protection against VACV via intraperitoneal injection. Additionally, we determined the complex structure of B6 and hMB668, revealing the structural feature of B6 and the epitope of hMB668. Collectively, our study provides two promising antibody candidates for the treatment of orthopoxvirus infections, including mpox.

The genus *Orthopoxvirus*, belonging to the *Poxviridae* family, comprises four human pathogenic species: monkeypox virus (MPXV), smallpox virus (variola virus, VARV), vaccina virus (VACV) and cowpox virus (CPXV)[1,2]. MPXV is the causative agent of the recent mpox epidemic, which was announced a Public Health Emergency of International Concern (PHEIC) by the World Health Organization (WHO) on 23 July 2022. This virus shows high similarity to other members in *Orthopoxvirus*[2,3]. Mpox is a zoonotic infection and was first identified in humans in 1970, and it mainly prevailed in the Central and West Africa

over the past half century[2,3]. However, since the first human case of mpox was reported in Europe in May 2022, mpox has emerged as a global health threat. As of 30 Novermber 2023, a total of 92,738 laboratory confirmed cases, including 171 deaths, have been reported in 116 countries and regions (https://worldhealthorg.shinyapps.io/mpx_global/).

Previous data indicated that the first-generation smallpox vaccine could provide 85% protection against MPXV[4], but the vaccination had been discontinued since the eradication of smallpox in 1980s. Two

[1]CAS Key Laboratory of Pathogen Microbiology and Immunology, Institute of Microbiology, Chinese Academy of Sciences, Beijing, China. [2]College of Veterinary Medicine, Shanxi Agricultural University, Jinzhong, Shanxi, China. [3]Institute of Physical Science and Information, Anhui University, Hefei, Anhui, China. [4]School of Life Sciences, Hebei University, Baoding, Hebei, China. [5]National Key Laboratory of Intelligent Tracking and Forecasting for infectious Diseases, Beijing Ditan Hospital, Capital Medical University, Beijing, China. [6]College of Future Technology, Peking University, Beijing, China. [7]University of Chinese Academy of Sciences, Beijing, China. [8]These authors contributed equally: Runchu Zhao, Lili Wu, Junqing Sun, Dezhi Liu. ✉e-mail: wangqihui@im.ac.cn

vaccines are approved by the US Food and Drug Administration (FDA) for pre-exposure vaccination against orthopoxviruses including MPXV: a second-generation attenuated replicating VACV vaccine, ACAM2000, and a third-generation attenuated nonreplicating vaccine based on modified vaccinia Ankara (MVA)[5,6]. However, ACAM2000 may cause side effects in certain patients such as those with eczema[7], and MVA needs further study on its effectiveness against mpox. Recently, real-world studies reported that MVA vaccination induced anti-MPXV neutralizing antibodies in young individuals, with seroprevalence of 52%[8] and 63%[9], respectively. The vaccine effectiveness of one dose of MVA was approximately 80% in male individuals or individuals at high-risk of mpox[10–12]. Additionally, LC16, a third-generation attenuated minimally-replicating vaccine developed and licensed for smallpox use in Japan[13], was also recommended by WHO for prophylaxis of mpox, however, its efficacy in humans against mpox needs further investigations. Although several therapeutics developed for smallpox, such as tecovirimat and brincidofovir, are available for mpox, their effectiveness for mpox is based on preclinical evidence with little clinical data from humans. Notably, vaccinia immune globulin (VIG) has been used for prevention and treatment of smallpox and vaccine-related complications, and has exhibited cross-neutralizing activity against MPXV in rhesus macaques[14]. However, neither VIG nor other monoclonal antibodies (MAbs) have yet been evaluated for their efficacy against mpox in humans. Therefore, there is an urgent need to develop effective countermeasures to combat mpox.

MAbs are efficient therapeutics against virus infections, as exemplified during the coronavirus disease 2019 (COVID-19) where several neutralizing MAbs have been authorized for emergence use[15–17]. Using VACV as the study model, multiple neutralizing MAbs targeting various antigens, including intracellular mature virus (IMV) surface proteins L1, A27, H3 and D8 and extracellular enveloped virus (EEV) surface proteins B5 and A33, have been identified[18]. Notably, antibodies targeting the B5 are mainly responsible for the EEV neutralizing capacity of human VIG, which demonstrates the prominent role of B5 as a target of EEV-neutralizing activity of human antibodies[19].

B5, a type I transmembrane glycoprotein, homologous to B6 in MPXV, is highly conserved among orthopoxvirus (Supplementary Fig. 1a, b)[20]. Apart from the stalk region, the extracellular domain of B5 contains four short consensus repeat (SCR) domains (Supplementary Fig. 1c), like those found in complement control proteins[20,21], but there is no evidence that B5 has complement regulatory activity[22]. Studies have shown that B5 plays crucial roles in various stages of viral infection. It is required for efficient wrapping of IMV, plaque size and virus virulence[23]. During EEV infection, B5 is required for glycosaminoglycan-mediated disruption of the EEV outer membrane, facilitating fusion of the inner membrane[24,25]. After infection, B5 is required on EEV to induce the formation of actin tails, thereby promoting accelerated spread to uninfected cells[26]. Previous studies have shown that polyclonal antibodies and MAbs from mouse, rat, chimpanzee, and human targeting B5 inhibited EEV infections or comet formation[18,27–32], and several, such as chimpanzee-human fusion MAb 8AH8AL, rat MAb 19C2 and human MAb h101, have been shown to provide protection in a mouse model[28,29,31]. Epitope-mapping studies have identified two major neutralizing sites on B5, located in SCR1-SCR2 and the stalk of B5[28,30]. However, the precise epitope information of these antibodies and other potential neutralizing epitopes on B5, as well as the structural characterization of B5, remain elusive.

Here, we focus on the incompletely characterized B5 to develop valuable antibody therapeutics for the treatment of orthopoxvirus infections. We initially evaluated the binding IgG antibodies to B6 protein of MPXV in plasma from individuals born before 1981 and further isolated two noncompeting MAbs, hMB621 and hMB668, targeting B6 from the individual with the highest binding antibody level. In vitro neutralizing assays indicated that the two MAbs (both generated as IgG1 subtypes) showed potent activities against VACV infections in the presence of complement. Moreover, in an intranasal VACV mouse challenge model, the two MAbs exhibited effective protection when given via intraperitoneal (i.p.) injection. Particularly, we also revealed the structural characterization of SCR1 and SCR2 domains of B6 and the epitope information of hMB668 by using cryo-electron microscopy (cryo-EM).

## Results

### Binding IgG antibodies to MPXV B6 in individuals born before 1981

With the global eradication of smallpox, China ended the smallpox vaccination around 1981. Therefore, we selected individuals born before 1981 to evaluate the preexisting cross-reactive antibody level to MPXV B6. 30 plasma samples from individuals born before 1981 (>40) and 20 plasma samples from individuals born after 1981 (<40) were collected in 2021 (Supplementary data 1). The former group included three age subgroups: 41-50, 51-60, and >60, each comprising 10 samples. We initially compared the binding IgG responses in plasma from the individuals born before and after 1981 to VACV B5 and MPXV B6. The result showed that VACV B5-binding IgG antibodies in plasma from individuals born before 1981 were significantly higher than those in plasma from individuals born after 1981, regardless of 1:100, 1:1000, or 1:10000 dilution (Fig. 1a). Similarly, IgG responses to MPXV B6 in the two age groups (>40 and <40) showed similar features as those observed for VACV B5, and there was no significant difference between plasma from individuals born before 1981, whether binding to VACV B5 or to MPXV B6 (Fig. 1a).

We further analyzed the IgG antibody responses in plasma at 1:100 dilution between three age subgroups. The results revealed that VACV B5- or MPXV B6-binding IgG antibodies showed significant difference between the individuals aged 41–50 and those below 40, as well as between the individuals aged 51–60 and those below 40 (Fig. 1b). Notably, no significant difference was observed between the individuals aged >60 and those below 40, which may be attributed to the weakening of immunity with age (Fig. 1b).

### Isolation and binding characterization of MPXV B6-targeting antibodies

To further investigate antibody characterization and develop valuable antibody agents targeting B6, we isolated the B6-specific memory B cells from the donor with the highest binding IgG to the B6, using a previous strategy[33]. Two MAbs hMB621 and hMB668, from different germline genes were identified, and their complementary determining regions (CDRs) in both heavy (H) and light (L) chains exhibited high somatic hypermutation (Supplementary Fig. 2). Surface plasmon resonance (SPR) results indicated that the hMB621 and hMB668 showed nanomolar binding affinities to B6, with equilibrium dissociation constant ($K_D$) values of 1.9 and 6.2 nM, respectively (Fig. 2a). Competition-binding assay revealed that the hMB621 and hMB668 did not compete for binding to B6 (Fig. 2b), indicating that they recognize two distinct epitopes on B6. We also compared the binding epitopes of the two MAbs with a previously reported MAb 8AH8AL (referred to as 8A here) identified from chimpanzee[28] and found that hMB668, but not hMB621, could compete with 8A (Supplementary Fig. 3). This result indicated that hMB668 may also recognize residues 20–130 of B6 like 8A, while hMB621 recognizes a non-overlapping epitope.

Due to the high conservation of B6 orthologs among orthopoxvirus (Supplementary Fig. 1a, b), we further evaluated the broad binding potencies of hMB621 and hMB668 to VACV, VARV and CPXV. Particularly, the expressed B6 protein used in this study (from clade II strain MPXV_USA_2022_MA001) has identical sequence to Clade I strain Zaire-96-I-16 (Supplementary Fig. 1a, b). SPR results indicated that both MAbs showed comparable abilities to bind VACV, VARV, and CPXV as to MPXV, except hMB668 to VARV and CPXV, the binding

a

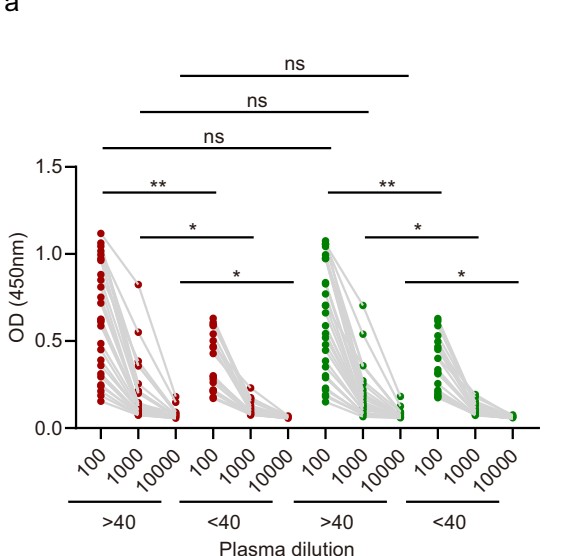

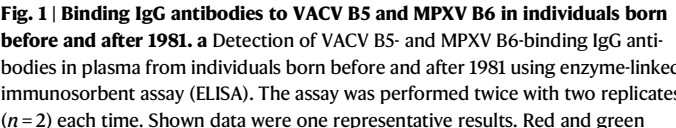

b

**Fig. 1 | Binding IgG antibodies to VACV B5 and MPXV B6 in individuals born before and after 1981. a** Detection of VACV B5- and MPXV B6-binding IgG antibodies in plasma from individuals born before and after 1981 using enzyme-linked immunosorbent assay (ELISA). The assay was performed twice with two replicates ($n = 2$) each time. Shown data were one representative results. Red and green

symbols represented IgG antibodies in plasma binding to VACV B5 and MPXV B6 proteins, respectively. **b** Analysis of IgG antibody responses in plasma at 1:100 dilution between different age subgroups. Statistical analysis in **a** and **b** were performed by using unpaired two-tailed t test. Source data are provided as a Source Data file.

abilities of which were 2–4 folds lower than that to MPXV or VACV (Fig. 2a).

## Complement-dependent neutralization of antibodies in vitro against VACV infection

Because B6 protein is specifically on EEV, we assessed the neutralizing activities of the hMB621 and hMB668 against EEV of VACV using a plaque reduction neutralization test (PRNT). The result indicated that both hMB621 and hMB668 exhibited potent neutralizing abilities against VACV in the presence of complement, with $PRNT_{50}$ values of 99.3 and 93.1 ng/mL, respectively (Fig. 3). As previously confirmed that B6-targeting neutralizing antibodies are complement dependent[31,32,34], the two MAbs showed poor potencies in the absence of complement.

## Protection of antibodies in vivo against VACV infection

The protection efficacies of the MAbs were evaluated in a BALB/c mouse model. Two doses of 10 mg/kg antibody were administrated via intraperitoneal (i.p.) injection 4 h before and 4 h after intranasal (i.n.) challenge of a lethal dose of VACV (Fig. 4a). Control mice were given an irrelevant MAb (against SARS-CoV-2) or PBS, and they exhibited persistent weight loss throughout the monitoring period, with exceeding 20% weight reduction at 5 days post-infection (dpi) (Fig. 4b). In contrast, hMB621 or hMB668 treated mice displayed weight increase at 4 dpi and all survived. All mice were euthanized at 6 dpi for viral load detection in the lung. As expected, compared to the control group, both hMB621 and hMB668 groups can significantly reduce viral loads in the lung, but the two groups showed no significance (Fig. 4c), which was consistent with their neutralizing potencies in vitro (Fig. 3). These results indicate that the two MAbs can effectively treat VACV infections in this mouse model.

## Molecular basis of antibodies binding to MPXV B6

To determine the binding epitopes of hMB621 and hMB668, we prepared extracellular full-length B6 (T20-H279) and three B6 fragments generated by C-terminal deletions: SCR1-2-3-4 (T20-N241), SCR1-2-3 (T20-K185) and SCR1-2 (T20-E129). ELISA results demonstrated that hMB621 showed a significant decrease in binding to SCR1-2-3 and an even further decrease to SCR1-2, compared to its binding to SCR1-2-3-4

and full-length B6 (Fig. 5a). On the other hand, hMB668 displayed comparable binding to all tested B6 fragments (Fig. 5a). The findings indicated that hMB621 probably binds to SCR3-4, while hMB668 binds to SCR1-2, which is consistent with their noncompetition-binding result (Fig. 2b).

To further elucidate the epitope information of the two MAbs, we prepared the complex of extracellular full-length B6 with hMB668 Fab and subjected it to cryo-electron microscope (cryo-EM). A 3.46 Å resolution of cryo-EM map was obtained (Fig. 5b; Supplementary Fig. 4, and Table 1). However, in the map, we only observed SCR1-2 of B6 and hMB668 Fab, despite extracellular full-length B6 was used. Like previously reported SCR structures in human complement activation (RCA) family proteins, including factor H (FH)[35], membrane cofactor protein (MCP)[36], C4b binding protein (C4BP)[37], decay-accelerating factor (DAF)[38] and complement receptor 1 (CR1)[39], as well as viral RCA proteins such as VCP (VACV complement control protein)[40], the SCR1 and SCR2 of B6 mainly folds into β sheets connected by intermediate linker, while the SCR2 additionally contained an α helix (Fig. 5c; Supplementary Fig. 5). Two N-linked glycosylation sites, N94 and N120, were also observed in SCR2. We then compared B6 with the complement regulator domains of human RCA family proteins, including SCR1-4 of FH, SCR1-4 of MCP, SCR1-4 of C4BP, SCR2-4 of DAF, and SCR1-4 of CR1, as well as SCR1-4 of VCP of VACV and SCR1-3 of MPXV inhibitor of complement enzymes (MOPICE) (Supplementary Fig. 5). Sequence and structure alignments indicated that there were obvious variations between B6 and these RCA proteins (Supplementary Fig. 5a–g).

When superimposing the SCR1-2 of B6 in the complex structure onto the predicted B6 model, a certain deviation is observed (Fig. 5d). However, when comparing the SCR1 and SCR2 in the two models individually, the two domains align well (Supplementary Fig. 5h, i). These findings suggest that the SCR1 or SCR2 in the predicted model resembles that in the solved structure. Nevertheless, the structural deviation observed when SCR1 and SCR2 as a whole could be attributed to the high flexibility of the loop between SCR1 and SCR2[41], or potentially influenced by the binding of hMB668. Notably, we also noticed that the H chain of hMB668 showed a steric clash with SCR3 and stalk regions of B6 (Fig. 5d), implying that the flexibility of the loop

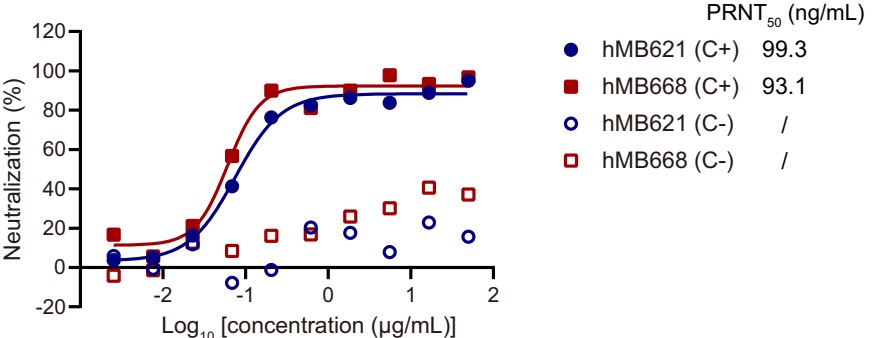

**Fig. 2 | Binding characterization of hMB621 and hMB668 to MPXV B6 and its orthologs in VACV, VARV and CPXV. a** Binding affinities of hMB621 and hMB668 to MPXV B6 and its orthologs in VACV, VARV and CPXV measured by surface plasmon resonance (SPR). The assay was repeated three times. Shown data were the mean ± standard deviation (SD) of three independent experiments. Shown curves were one representative results of three independent experiments. **b** Competitive binding of hMB621 and hMB668 on MPXV B6 measured by bio-layer interferometry (BLI). Biotinylated MPXV B6 protein was first immobilized on the streptavidin (SA) sensor. Then, the sensor was exposed to the first antibody (Ab1), followed by exposure to the second antibody (Ab2) in the presence of the Ab1. The assay was performed twice. Shown data are one representative result. Source data are provided as a Source Data file.

**Fig. 3 | Neutralizing activities of hMB621 and hMB668 against VACV.** Neutralizing potencies of hMB621 and hMB668 against EEV of VACV with (C + ) or without rabbit complement (C-) were tested using a plaque reduction neutralization test (PRNT). The assay was performed three times with one well at each antibody concentration ($n = 1$). Representative curves of three independent experiments were shown. $PRNT_{50}$ values were the mean of three independent experiments. Source data are provided as a Source Data file.

between SCR2 and SCR3 that permits the binding of hMB668. This flexibility possibly contributes to the lack of SCR3-4 and stalk regions in the map.

The interaction analysis between hMB668 and B6 revealed that the H chain of hMB668 predominantly engaged with SCR2 of B6, while the L chain of hMB668 interacted with SCR1 and SCR2 of B6, with a preference for SCR1 (Fig. 5e–g). Regarding the H chain of hMB668, all three CDRs are involved in the interaction with B6 (Fig. 5f). While for the L chain of hMB668, CDR1, and CDR2, but not CDR3, participates in interaction with B6 (Fig. 5g). Notably, among the binding sites of hMB668 on B6, only H53 is not conserved in VARV and CPXV, both of which are a tyrosine (Y) at this site (Supplementary Fig. 1a). According

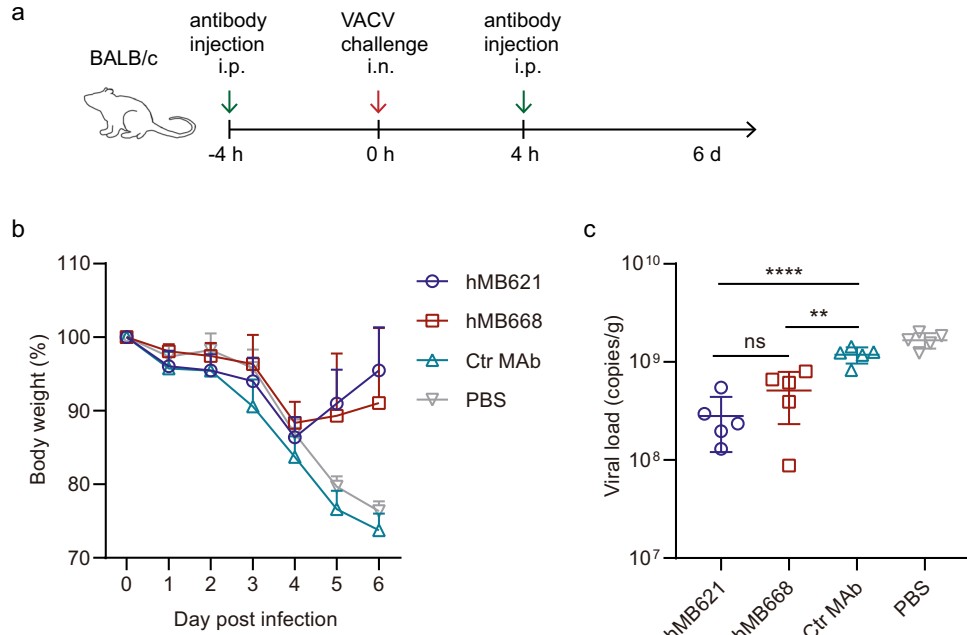

**Fig. 4 | Protection efficacy of hMB621 and hMB668 against VACV in vivo.**
**a** BALB/c mice (*n* = 5) were treated intraperitoneally (i.p.) with 10 mg/kg indicated antibody or PBS 4 h before and 4 h after intranasal (i.n.) challenge with a lethal dose of VACV WR strain. The experiment was performed once. **b** Body weight changes of BALB/c mice. Ctr MAb, a MAb against SARS-CoV−2. **c** Viral titers in lung 6 days post-infection (dpi) were detected by using quantitative real-time PCR. Statistical analysis was performed by using unpaired two-tailed t test. Source data are provided as a Source Data file.

to the structural analysis, we speculated that the H53Y change in VARV and CPXV possibly resulted in the slightly decreased binding of hMB668 by increasing steric clash with Y54 in hMB668 L chain (Fig. 5g). As expected, SPR assays indicated that the abilities of hMB668 binding to B6 orthologs of VARV and CPXV with Y53H mutation improved to the level of hMB668 to MPXV or VACV (Figs. 5h, i and 2).

## Discussion

The current outbreak of mpox is the largest and most widespread in the history of mpox, raising concerns about a new global public health threat. Although mpox is self-limiting in most cases, severe clinical manifestations and complications have been reported (https://www.who.int/news-room/fact-sheets/detail/monkeypox). Therefore, there is an urgent need for therapeutic development to combat the mpox. Extensive antibody studies have been conducted for VACV, however, the research on MPXV remains limited. In this study, we focus on B6 of MPXV, the crucial immunogen on EEV surface, to develop valuable therapeutic antibodies against MPXV.

We initially evaluated the level of preexisting antibodies to B6 in the individuals who were smallpox-vaccinated (born before 1981) though had no historic vaccination records. ELISA results revealed that the binding antibody level to MPXV B6 in people born before 1981 was similar to that observed for VACV B5, indicating the high cross-reactivity of B6-binding antibodies between VACV and MPXV and suggesting longevity of antibodies induced by smallpox vaccination[42]. However, in individuals aged >60, the binding antibody level was lower than that in the people aged 51-60, indicating a decline in antibody level over time. Notably, in the people aged 41-50, the binding antibody level was lower than that in the people aged 51-60, which could be attributed to some individuals being unvaccinated due to the discontinuation of smallpox vaccination around 1981.

To further explore the B6-targeting antibody characterization in smallpox-vaccinated individuals and develop potential antibody therapeutics, we isolated two neutralizing MAbs, hMB621 and hMB668, from the individual who showed the highest binding antibody level to

B6 among tested. Since we could not work with MPXV, we initially assessed the antiviral activities of these two MAbs against VACV both in vitro and in vivo and found that both MAbs displayed similarly potent antiviral potencies. Notably, hMB621 and hMB668 recognized two distinct epitopes on B6. hMB621 targeted the SCR3-SCR4, while hMB668 recognized the SCR1-SCR2. This finding suggested that, in addition to the previously reported neutralizing epitopes in the SCR1-SCR2 and stalk of B6[30], the SCR3-SCR4 of B6 also serves as a neutralizing epitope. However, when we combined the two MAbs, we observed no obvious synergistic effect against VACV (Supplementary Fig. 6). In addition, despite hMB668 exhibiting a slight reduction in binding to VARV and CPXV with the H53Y change compared to MPXV and VARV, both hMB668 and hMB621 still showed broad cross-binding activities to these viruses due to the high conservation of B6. This result suggested that they could also neutralize infections of MPXV, VARV, and CPXV, making them promising candidates for antiviral therapeutics. Particularly, their antiviral effectiveness against MPXV infections should be evaluated in the future. Moreover, previous data indicated that a mixture of neutralizing antibodies targeting B6 and other antigens, including A35, M1, and A29, could achieve the broad cross-neutralization against orthopoxvirus[18]. Therefore, further studies are needed to identify MAbs targeting these antigens, to make MAbs cocktails that recognize both EEV and IMV, which may exhibit better antiviral potency.

Extensive studies have been carried out for the function of VACV B5, but its three-dimensional structure remains elusive. Here, we have successfully obtained the cryo-EM structure of SCR1 and SCR2 domains of B6 and preliminarily revealed the epitope of a neutralizing antibody for the first time. However, the resolution of the structure is limited, warranting further optimization in future studies. Additionally, only the SCR1 and SCR2 domains were observed in the structure, indicating that the flexibility of the linkers between SCRs might be a possible reason leading to the lack of domains in the map as well as for the delay in revealing the complete structure of B6. Thus, we propose that utilizing multiple antibodies targeting distinct epitopes may help stabilize the conformation of B6, thus facilitating the structural

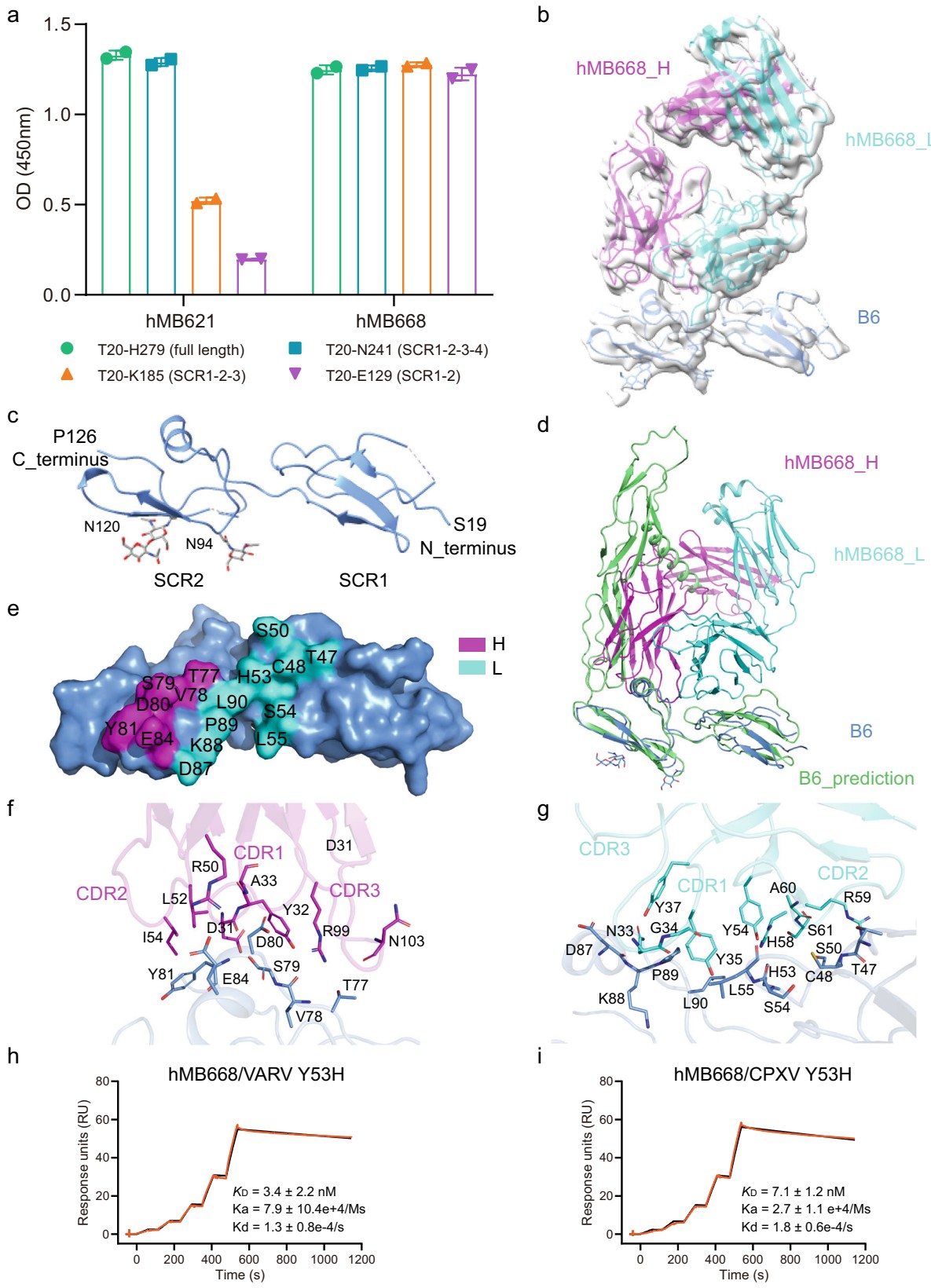

determination of the protein. However, despite we prepared the complex of B6 with hMB621 and hMB668, we have not yet determined the ternary complex structure, which requires further exploration in the future. In addition, multiple antibodies targeting B6 should also be developed.

Although the WHO has declared the mpox epidemic no longer a PHEIC, mpox confirmed cases are still occurring. Notably, most cases in the current mpox outbreak are men who have sex with men (MSM), and the tightly connected sexual networks within this community might contribute to the rapid transmission of mpox[43]. Additionally,

**Fig. 5 | Structural characterization of hMB668 binding to MPXV B6. a** Detection of hMB621 and hMB668 binding to different fragments of MPXV B6 proteins using ELISA. The assay was repeated twice with two replicates (*n* = 2) each time. One representative result was shown. **b** Structural model and cryo-EM density of hMB668 Fab binding to B6. The heavy (H) chain and light (L) chain of hMB668 and B6 were colored in magenta, cyan and light blue, respectively. **c** A cartoon representation of the SCR1 and SCR2 domains of B6 structure. **d** Superimposition of the SCR1 and SCR2 in predicted B6 structure onto the structure of B6 in complex with hMB668. **e** The footprint of hMB668 on B6. **f, g** The interaction analysis between H chain of hMB668 (**f**) and B6 and between L chain of hMB668 (**g**) and B6. The residues involved in the interaction were shown as sticks. **h, i** Binding affinities of hMB668 to B6 orthologs in VARV (**h**) and CPXV (**i**) with Y53H mutation measured by SPR. The assay was repeated three times. Shown data were the mean ± SD of three independent experiments. Shown curves were one representative results of three independent experiments. Source data are provided as a Source Data file.

several reports from the US and Europe suggested that around 40% and up to 90% of mpox cases occurred in people with human immunodeficiency virus (HIV)[44], resulting in more protracted and severe symptoms than those without HIV[45,46]. Therefore, there is an urgent need for the development of efficient therapeutics against MPXV, particularly for individuals who are living with HIV and exhibit poor immune response.

In summary, we isolated two potent neutralizing MAbs against MPXV B6 from a smallpox-vaccinated individual. Both MAbs can cross-react to several orthopoxviruses including MPXV, VACV, VARV and CPXV and exhibited efficient antiviral activities against VACV infections in vitro and in vivo, indicating their potential as broad therapeutics for treatment of mpox and other related diseases.

## Methods
### Cells and viruses
Human embryonic kidney (HEK) Expi293F cells (Sino Biological) were cultured at 37 °C in SMM 293-TII expression medium with 5% $CO_2$ in a shaking incubator (150 rpm). HEK293T cells (ATCC, CRL-3216), Vero cells (ATCC, CL-81), and HeLa cells (ATCC, CCL-2) were cultured at 37 °C in Dulbecco's modified Eagle medium (DMEM) (Gibco, C11995500BT) supplemented with 10% fetal bovine serum (FBS) (Gibco, 10437-028). The initial stock of VACV Western Reserve (VACV-WR) was obtained from Prof. Min Fang from Institute of Microbiology, Chinese Academy of Sciences (CAS).

### Isolation of peripheral blood mononuclear cells (PBMCs) and plasma
Blood samples were collected from fifty individuals (Supplementary data 1), including 26 male participants and 24 female participants (aged 16-73), who had recovered from COVID-19 at Beijing Ditan Hospital, China. Written informed consent was obtained from all participants. The study received approval from the Research Ethics Committee of the Institute of Microbiology, CAS (APIMCAS2021149). PBMCs were isolated following the manufacturer's instructions (Dakewe Biotech, 7922112) and stored in liquid nitrogen before use, and plasma were storage at −80 °C until use.

### Memory B cells isolation and recombinant monoclonal antibody production
Briefly, thawed PBMCs were incubated with His-tagged MPXV B6 at 400 nM before staining with anti-CD3/PE-Cy5 (BD Pharmingen™, Cat No. 555334, Clone No. UCHT1, Dilution: 1:20), anti-CD16/PE-Cy5 (BD Pharmingen™, Cat No. 555408, Clone No. 3G8, Dilution: 1:20), anti-CD235a/ PE-Cy5 (BD Pharmingen™, Cat No. 559944, Clone No. GA-R2, Dilution: 1:20), anti-CD19/APC-Cy7 (BD Pharmingen™, Cat No. 557791, Clone No. SJ25C1, Dilution: 1:20), anti-CD27/Pacific Blue (Biolegend, Cat No. 302822, Clone No. O323, Dilution: 1:50), anti-IgG/FITC (BD Pharmingen™, Cat No. 555786, Clone No. G18-145, Dilution: 1:20), anti-His/PE (Miltenyi Biotec, Cat No. 130-120-718, Clone No. GG11-8F3.5.1, Dilution: 1:10), as previously reported[33]. Antigen-specific memory B cells (CD3⁻, CD16⁻, CD235a⁻, CD19⁺, CD27⁺, IgG⁺, and His⁺) were sorted into 96-well PCR plates containing lysis buffer, followed by RT-PCR and nested PCR to amplify the genes encoding variable regions of heavy and light chains of antibodies. Subsequently, the genes were cloned

into pCAGGS vector containing the leader sequence (METDTLLLWVLLLWVPGSTGD) and constant region of human IgG1 (for heavy chain), Igκ (for κ chain) or Igλ (for λ chain) to generate antibody expression plasmids.

### Protein expression and purification
The optimized sequences of MPXV B6 (accession no. URK20605.1), VACV B5 (accession no. YP_233069.1), VARV B7 (accession no. NP_042219.1) and CPXV CPXV199 (accession no. NP_619980.1) were fused with N-terminal mouse Igκ signal peptide and C-terminal 6×His tag and then cloned into pCAGGS expression vector, respectively. The pCAGGS plasmids were transiently transfected into Expi293F cells. After 4 days, the supernatants were collected, and the soluble proteins were purified by Ni affinity chromatography with a HisTrap™ HP 5-ml column (GE Healthcare). The samples were further purified via gel filtration chromatography with a HiLoad 16/600 Superdex™ 200 Pg column (GE Healthcare) in phosphate-buffered saline (PBS).

The pET-21a plasmids containing MPXV B6 (T20-H279), B6 (T20-N241), B6 (T20-K185) or B6 (T20-E129) were transformed into *Escherichia coli* (*E. coli*) strain BL21 (DE3) for protein expression. After induction with 1 mM IPTG for 8 h at 16 °C, B6 (T20-H279) and its truncated forms were overexpressed as inclusion bodies and refolded, as previously reported[47]. After refolding, the proteins were concentrated and exchanged in PBS buffer. Subsequently, the proteins were further purified by gel filtration using a HiLoad Superdex™ 75 Pg (GE Healthcare).

The monoclonal antibodies were expressed in Expi293F cells by transient transfection. Supernatants containing monoclonal antibodies were collected and passed through a Protein A affinity column (GE Healthcare) and further purified using HiLoad 16/600 Superdex™ 200 Pg (GE Healthcare) in PBS.

### Enzyme-linked immunosorbent assay
The binding IgG antibodies in the plasma to VACV B5 and MPXV B6 were determined by an enzyme-linked immunosorbent assay (ELISA). Purified VACV B5 and MPXV B6 were immobilized at 200 ng per well in 0.05 M carbonate-bicarbonate buffer (pH 9.6) on 96-well plates (Corning, USA) and left to incubate at 4 °C overnight. After blocking with 5% skim milk in PBST buffer (1.8 mM $KH_2PO_4$, 10 mM $Na_2HPO_4$ (pH 7.4), 137 mM NaCl, 2.7 mM KCl, and 0.005% (v/v) Tween 20) at room temperature for 1 h, diluted plasma samples (at 1:100, 1:1000 and 1:10,000 dilutions) were added and incubated at room temperature for 1 h. Subsequently, the plates were incubated with goat anti-human IgG-HRP antibody (ZSGB-Bio, ZB-2304) and developed with 3,3′,5,5′-tetramethylbenzidine (TMB) (Beyotime, P0209) substrate. The reactions were stopped with 2M sulfuric acid and the absorbance at 450 nm was measured. The results were analyzed by using GraphPad Prism 8.0.2.

### BLI assay
For the competitive binding experiment between hMB621 and hMB668, biotinylated MPXV B6 protein (10 μg/mL) was first immobilized on the streptavidin (SA) sensor (GE Healthcare). Then, the sensor was exposed to the first antibody (50 μg/mL) for 500 s, followed by exposure to the second antibody (50 μg/mL) in the presence of the

same concentration of the first antibody for 500 s. All experiments were conducted using a Biolayer interferometry (BLI) system with an Octet RED96 biosensor (FortéBio) at room temperature.

## SPR analysis

The antibodies (0.5 µg/mL) were first captured on flow cell 2 of the Protein A sensor chip (GE Healthcare) to achieve approximately 500 response units (RU). Flow cell 1 was utilized as the negative control. Then, serially diluted MPXV B6, VACV B5, VARV B7, or CPXV CPXV199 proteins were flowed over the chip in PBST buffer. The RU were measured with a BIAcore 8K (Cytiva) at 25 °C in a single cycle mode. The sensor chip was regenerated with 10 mM glycine-HCl (pH 1.5). The equilibrium dissociation constants ($K_D$) for each pair of interactions were calculated using BIAcore® 8K Evaluation Software (Cytiva) by fitting the data to a 1:1 Langmuir binding model.

## VACV EEV preparations

As previously described[32], enveloped extracellular virion (EEV) of VACV-WR were prepared using monolayers of HeLa cells in DMEM supplemented with 2% FBS (heat-inactivated at 56 °C for 30 min) in T75 flasks at 90% confluence. The cells were infected at a multiplicity of infection (MOI) of 0.5. The medium containing the EEV form was harvested at 48 h and were centrifuged (450×g for 8 min at 4 °C) to remove cells. The clarified supernatant was stored at 4 °C and used within 2 weeks. The titer of the EEV was determined in the presence of an intracellular mature virion (IMV)-neutralizing anti-L1 antibody (7D11)[48] on Vero cells (-2 × 10^5 PFU/mL).

## VACV EEV neutralization assay

The neutralizing activities of hMB621 and hMB668 were determined using the EEV form of VACV-WR in a plaque reduction neutralization (PRNT) assay. Neutralization of VACV EEV was performed using 10% baby rabbit complement (Cedarlane, CL3441-R) and in the presence of an IMV-neutralizing anti-L1 antibody (7D11) at a concentration of 50 µg/mL. Briefly, 0.2 mL DMEM containing -100 PFU of EEV, 7D11 and complement was incubated with 0.2 mL of serial three-fold dilutions of antibody for 2 h at 37 °C. Monolayers of Vero cells in 12-well plates were then infected with incubated antibody-EEV-complement mixture for 1 h at 37 °C. After infection, a semisolid 2% methylcellulose of Earle's basal minimal essential medium overlay was added to wells. After plates were incubated 40 h at 37 °C, Vero cells were fixed with 4% paraformaldehyde (Solarbio, P1110) at room temperature for 1 h and then stained with a 1% crystal violet solution for 1 h. At last, the plates were rinsed with water, and the plaques were counted. The half-maximal inhibitory concentration ($IC_{50}$) values were determined using GraphPad Prism 8.0.2.

## Animal protection experiments

All animal experiments were conducted in compliance with the guidelines and regulations of animal welfare and were approved by the Animal Ethics Committee of the Institute of Microbiology, CAS (APIMCAS2022124). The experiments were performed in an Animal Biosafety Level 2 facility under the condition of 12 h light and dark cycle, temperature of 20-25°C, humidity of 40-70%.

In all, 6- to 8-week-old female BALB/c mice (Vital River) were given antibodies intraperitoneally (i.p.) 4 h before and 4 h after a lethal dose (5 × 10^4 PFU) of VACV inoculation via the intranasal route (i.n.). The mice were monitored daily for weight changes to evaluate the protective effect of the antibodies. Mice that exhibited weight loss of over 25% of their initial weight were euthanized in accordance with ethical guidelines.

To evaluate viral loads, the lungs of the mice were collected, weighed and then homogenized into 500 µL serum-free RPMI 1640 (Gibco, C11875500BT) using a tissue homogenizer (NZK) at 4000 rpm

for 60 s (grinding for 10 s, pausing for 10 s and repeating 6 times). The supernatants were isolated for viral RNA extraction using the QIAamp MinElute Virus Spin Kit (Qiagen, 57704) as template of quantitative real-time PCR (qPCR).

The detection of VACV was performed with NovoStart® Probe qPCR SuperMix (UDG) Kit (Novoprotein, E106-01S) on the ABI QuantStudio 3 Real-Time PCR system with the following conditions: 50 °C for 2 min and 95 °C for 2 min, followed by 45 cycles of amplification at 95 °C for 10 s and 60 °C for 40 s. The primers and TaqMan® Minor Groove Binding (MGB) probe used in the qPCR assay targeted the *E9L* gene of VACV-WR, with the following sequences: Forward primer: 5'-CGGCTAAGAGTTGCACATCCA-3', Reverse primer: 5'-CTCTGCTCCATTTAGTACCGATTCT-3', TaqMan® MGB probe: AGGACGTAGAATGATCTTGTA.

## Cryo-EM sample preparation and data acquisition

To prepare the cryo sample, the MPXV B6-hMB668 complex sample was vitrified using a Vitrobot Mark IV (ThermoFisher Scientific) plunge freezing device. The sample (4.0 µL, 0.1 mg/ml) was applied to graphene oxide supported UltrAuFoil (GO dataset) grids. Grids above were then blotted using different conditions (blot time 2 s and blot force -2; blot time 2 s and blot force -4) at a temperature of 4 °C and a humidity level of >99% and plunge frozen into liquid ethane.

The prepared grids were transferred to a 300 kV Titan Krios transmission electron microscope equipped with Gatan K3 detector and GIF Quantum energy filter. Movies were collected at ×105,000 magnification with a calibrated pixel size of 0.69 Å over a defocus range of −1.0 µm to −2.0 µm in super resolution counting mode with a total dose of 60 e⁻/Å² using EPU (ThermoFisher Scientific) automated acquisition software.

## Image processing

The detailed data processing workflow is summarized in Supplementary Fig. 4. All the raw dose-fractionated images stacks were 2× binned, aligned, dose-weighted, and summed using MotionCor2[49]. The contrast transfer function (CTF) estimation, particle picking and extraction, 2D classification, ab initio model generation, 3D refinements were performed in cryoSPARC v.4.2.1[50].

For the MPXV B6-hMB668 complex, a total of 23,530 micrographs were collected for this dataset. We picked out particles using blob-pick procedure of cryoSPARC from 1000 micrographs, and then these particles were subjected to 2D classification. After three rounds of 2D classification, we selected good particles in different views for Topaz training and then generated the Topaz model. Then we applied the Topaz[51] procedure to select particles against entire micrographs. The 1,555,386 initial particles were picked and extracted with the box size of 360 pixels from 23,530 micrographs. After the extensive 2D classification, approximately 806,766 good particles were selected to generate the initial models and two rounds of Heterogeneous Refinement and resulted to three distinct volumes. The one dominant class containing 34.9% of total particles was identified, which displayed clear features of secondary structural elements. To improve the local resolution of the MPXV B6 subunit, we generated a mask around B6 and performed 3D classification without alignment to pick out good particles with whole B6 density using Relion v.3.1.0[52] coupled with cryoSPARC v.4.2.1. One rounds of 3D classification reduced the number of particles to 208,637 leading to an EM density map at an overall 3.46 Å resolution.

## Model building and structure refinement

The MPXV B6 protein was initially predicated by AlphaFold2[53]. For the structure of the hMB668, the model (PDB: 4DN3) was rigidly docked into the density map using Chimera[54]. Mutation and manual adjustment were carried out with Coot v.0.9.3[55]. Glycans were added at N-linked glycosylation sites in Coot. Each residue was manually checked with the

chemical properties taken into consideration during model building. Several rounds of the real-space refinement in Phenix-1.20.1[56] and manually building in Coot and online sever (https://namdinator.au.dk/)[57] were performed until the final reliable models were obtained. Molprobity[58] was used to validate geometry and check structure quality. Statistics associated with data collection, 3D reconstruction, and model building were summarized in Supplementary Table 1. Figures were generated using ChimeraX[59] and PyMol (http://www.pymol.org).

## Declaration of Generative AI and AI-assisted technologies in the writing process
During the preparation of this manuscript, the authors used ChatGPT to polish the English language of the introduction, result and discussion parts of the manuscript, without changing the content. The authors take full responsibility for the content of the publication.

## Reporting summary
Further information on research design is available in the Nature Portfolio Reporting Summary linked to this article.

## Data availability
Cryo-EM density map and atomic coordinates have been deposited in the Electron Microscopy Data Bank and Protein Data Bank with the accession codes EMD-38613 and PDB: 8XS3, respectively. The sequences of hMB621 and hMB668 have been deposited in Genome Sequence Archive (https://ngdc.cncb.ac.cn/gsa-human/) with accession number HRA006123. Source data are provided with this paper.

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

## Acknowledgements

We thank Prof. Min Fang (Institute of Microbiology, Chinese Academy of Sciences) for providing the VACV Western Reserve (VACV-WR) virus. We thank the staff at the Cryo-Electron Microscope Center, Shanxi Academy of Advanced Research and Innovation, for their operation and maintenance of the equipment. This work was supported by the National Key R&D Program of China (2022YFC2303400 to Q.W. and 2022YFC260 4100 to L.W.), the National Natural Science Foundation of China (82225021 to Q.W.), and the Chinese Academy of Sciences (YSBR-010 and Y2022037 to Q.W.).

## Author contributions

Q.W. and G.F.G. initiated and coordinated the project. R.Z. and L.W. isolated the B6-specific antibodies, with H.W. providing the B6 protein and Z.C. and Y.X. providing the blood samples. R.Z. performed the ELISA, BLI, and SPR assays and prepared the hMB668 and B6 complex protein. P.H. prepared the cryo-EM samples and collected the cryo-EM data, and J.S. solved the cryo-EM structure with the help of Y.C. D.L., and R.Z. performed the neutralization and protection assays with the help of X.Q., Y.G., and Y.Z.; L.W., Q.W., and G.F.G. analyzed the data and wrote the manuscript.

## Competing interests

A patent (application number 2023108890941) was filed containing the hMB668 and hMB621 antibodies described in this study. Q.W., G.F.G., L.W., R.Z., and D.L. are the inventors. There are no restrictions on the publication of the data. The other authors declare that they have no competing interests.
