## [Peer Review File · Nature Communications]

REVIEWER COMMENTS

Reviewer #1 (Remarks to the Author):

The manuscript by Zhao et al describes two monoclonal antibodies (mAbs), hMB621 and hMB668, targeting distinct epitopes on the MPXV B6 protein. Using binding assays, the authors demonstrate that these mAbs can bind to B6 orthologs in vaccinia, variola, and cowpox viruses. These data are further complemented by neutralization and in vivo protection data using vaccinia. Finally, the authors determined the cryo-EM structure of B6 and hMB668, revealing the molecular basis of the neutralization and protection of the mAb targeting B6. It is this latter aspect of the work that I will focus my review on.

Given the public health threat posed by MPXV, the results of this study are important. Moreover, the manuscript is well-written, and the results are straightforward. However, there are serious issues with the cryo-EM data and its overinterpretation. As such, I cannot recommend the publication of this data in its current form as it would not stand up to the scrutiny of the structural biology community.

Major Concerns

1) The authors utilized DeepEMhancer, a deep learning approach designed for automatic post-processing of cryo-EM maps, to modify their map. While such methods are not inherently frowned upon in the cryo-EM community, in this particular instance, the resulting modified map introduces features that are not present in the unmodified data. This essentially masks issues with the map, specifically its high anisotropy. This is evident through the following:

- The orientation distribution plot in Figure S4D.
- The 2D class averages in Figure S4B (see below). Only two unique views of the complex seem to be apparent.
- The unmodified sharpened map lacks features expected in a 3.4 Å resolution map.

2) Related to point 1, the authors have built an atomic model using the DeepEMhancer map and used this for refinement. Given that the AI-modified map has features that are not present in the

unmodified data, it's not clear to me if this model can be trusted. This probably explain the poor validation scores for the result pdb file compared to models of a similar resolution.

Minor issues

1) The FSC curve shown in Figure S4C never reaches zero. This is indicative of issues with the data, such as the presence of duplicate particles. <https://discuss.cryosparc.com/t/why-do-duplicate-particles-appear/12072>

2) The authors have performed single-particle image processing using particles with a pixel size of 0.68 Å/pixel. While not strictly incorrect, the resolution of the resulting map is so far away from Nyquist that using such a small pixel size makes no sense. In fact, by binning the data 1.5x, the improved signal-to-noise may improve alignment of particles during refinement.

Recommendations

Despite the major issues with the cryo-EM map and model, the overall interpretation of the structural data appears to be correct. The resolution of the (unmodified map) is good enough to fit the AlphaFold model and make an educated guess about the epitope-paratope residues. The way I see it, the authors have two options on how to proceed.

Option 1: Be more forthcoming with the limitations of the current data set.

- The authors should carry out orientation diagnostics of their data and include this as a supplementary figure. This analysis can be performed in the latest version of CryoSPARC.

<https://guide.cryosparc.com/processing-data/all-job-types-in-cryosparc/utilities/job-orientation-diagnostics>

- The authors should show example density for the modified and unmodified regions of the map with the fitted pdb.
- Remove domain-level RMSD values. Given the limited resolution, these values are not reliable. Description of movements between domains would be ok to describe at this resolution.
- I would strongly advise using unmodified maps for refinement. There are several nice tools for flexibly fitting models (such as your AlphaFold coordinates) into lower resolution maps (e.g. Namdinator <https://namdinator.au.dk/>).

Option 2: Obtain a higher-quality experimental structure.

- I appreciate this is probably not feasible given the time and effort required to generate a structure, but there are several ways to alleviate the strong preferred orientation seen in your data. Collecting a tilted data set would be the most straightforward. I can think of one example of a similar sized complex on graphene oxide that was determined from tilted data:
<https://www.nature.com/articles/s41467-023-38509-2#Sec18>

Reviewer #2 (Remarks to the Author):

This manuscript describes the isolation, expression, and characterization of two anti-B5 human monoclonal antibodies from B-cells from a previously vaccinated person. One (hMB668) competes with binding to mpox-B5 of a previously published monoclonal (MAb 8AH8AL, ref 26). A cryo-EM structure of hMB668 bound to the first two SCR's of mpox-B5 is also presented and data from that may explain the slightly lower binding of this MAb to the VARV- and CPXV-B5 proteins compared to the MPXV- and VACV-B5 proteins. Authors provide evidence that the other human MAb isolated (hMB621) bound to SCR3-SCR4, but Cryo-EM structure of this complex could not be obtained.

Both MAb showed EV neutralizing activity in the presence of complement and when given 4 hr before and 4 hr after challenge, showed protection of mice in an intranasal mouse challenge model with VACV.

B5 is known to be an important target of a protective immune response and this paper supports this and provides additional important characterization of B5 and human MAbs that may some day be used as part of a cocktail of monoclonal antibodies that could be used for the treatment or prevention of orthopoxvirus infections.

Major points for authors to address:

What subtype IgG are these MAbs?

Do these antibodies inhibit comet plaque formation?

Need to include the genebank names of the genes expressed. Is the sequence of those expressed proteins the ones shown in Supplementary figure 1A?

Minor points

1. Abstract, line 33. I do not think the data shows the molecular basis of the neutralization and protection of the MAb targeting the B6. At best the cryo EM shows the site of binding /interaction with B6.
2. Line 38/39. Would include the word complement since that is a key feature of the neutralization activity.
3. Lines 56. Statements not entirely correct. In U.S., the Jynneos (MVA) vaccine is approved as a vaccine to prevent mpox.
4. Lines 64 & 67. While there had been limited clinical data on its effectiveness of MVA to protect against mpox, since the outbreak there has been a lot more data. Authors should also cite more studies that show better effectiveness than the one study they cite.
5. Line 78. Instead of clinic, would say in humans.
6. Line 110. Would not call B5 a poorly characterized protein. This has been an extensively studied orthopoxvirus protein
7. Line 115. Would end sentence with “in the presence of complement.”
8. Line 116. Sentence should read, Moreover, in an intranasal VACV mouse challenge model, the two MAbs exhibited effective protection when given via intraperitoneal (i.p.) injection.
9. Line 170. Title should read: Neutralization of antibodies in vitro against VACV infection is complement dependent
10. Line 181. Would include what a 10 mg/kg dose represents in mice to better compare to published data in ref 25 which used humanized MAb at 90 ug of purified Mab.
11. Lines 190/191. Given the nature of the challenge model (likely non-lethal challenge dose and other factors), better today, data shows that these MAbs can effectively treat VACV infections in this mouse model.
12. Line 193. More correct title: Molecular basis of antibodies binding to MPXV B6
13. Line 282. Better to say, “Since we could not work with live MPXV ..”

14. Line 287. Since neutralization is complement dependent, not sure these can be called "neutralizing epitopes".
15. Line 293. Similar to the prior comment, not sure one can say neutralize.
16. Line 303. Instead of determined, might say obtained.
17. Line 316. Occurring instead of increasing
18. Figure 2B. legend should explain how competitive binding was done.
19. Figure 3 legend. Would indicate species of complement (rabbit)
20. Figure 4. How many times was the challenge experiment performed. Seems like it was just once with 5-mice per group?
21. Supplementary figure 1. Need to include gene bank name of proteins being used. Also given interest in Clade 1 and Clade 2 MPXV, would include a representative B5 sequence from each.
22. Supplementary Figure 5. None of the colors used in the figure are explained. For example, panel H and I. What is the SCR and what is the antibody?

REVIEWER COMMENTS

Reviewer #1 (Remarks to the Author):

The manuscript by Zhao et al describes two monoclonal antibodies (mAbs), hMB621 and hMB668, targeting distinct epitopes on the MPXV B6 protein. Using binding assays, the authors demonstrate that these mAbs can bind to B6 orthologs in vaccinia, variola, and cowpox viruses. These data are further complemented by neutralization and in vivo protection data using vaccinia. Finally, the authors determined the cryo-EM structure of B6 and hMB668, revealing the molecular basis of the neutralization and protection of the mAb targeting B6. It is this latter aspect of the work that I will focus my review on.

Given the public health threat posed by MPXV, the results of this study are important. Moreover, the manuscript is well-written, and the results are straightforward. However, there are serious issues with the cryo-EM data and its overinterpretation. As such, I cannot, recommend the publication of this data in its current form as it would not stand up to the scrutiny of the structural biology community.

Major Concerns

1) The authors utilized DeepEMhancer, a deep learning approach designed for automatic post-processing of cryo-EM maps, to modify their map. While such methods are not inherently frowned upon in the cryo-EM community, in this particular instance, the resulting modified map introduces features that are not present in the unmodified data. This essentially masks issues with the map, specifically its high anisotropy. This is evident through the following:

- The orientation distribution plot in Figure S4D.
- The 2D class averages in Figure S4B (see below). Only two unique views of the complex seem to be apparent.
- The unmodified sharpened map lacks features expected in a 3.4 Å resolution map.

Response: Thanks for your constructive suggestions. Following your suggestions, we reprocessed the cryo-EM data by removing the duplicates and finally acquired an unmodified map at a resolution of 3.46 Å. This map was used for refinement. We also performed orientation diagnostics and prepared the new Figure 5 and Supplementary Figure 4, shown as follows.

Figure 5

Supplementary Figure 4

2) Related to point 1, the authors have built an atomic model using the DeepEMhancer map and used this for refinement. Given that the AI-modified map has features that are not present in the unmodified data, it's not clear to me if this model can be trusted. This probably explain the poor validation scores for the result pdb file compared to models of a similar resolution.

Response: Using the unmodified map for refinement resulted in an improved validation score of 16 for the PDB file, compared to the previous score of 26.

Minor issues

1) The FSC curve shown in Figure S4C never reaches zero. This is indicative of issues with the data, such as the presence of duplicate particles. <https://discuss.cryosparc.com/t/why-do-duplicate-particles-appear/12072>

Response: By reprocessing the cryo-EM data, including the removal of duplicates, the resulting FSC curve reached zero, as shown in the new Supplementary Figure 4c.

2) The authors have performed single-particle image processing using particles with a pixel size of 0.68 Å/pixel. While not strictly incorrect, the resolution of the resulting map is so far away from Nyquist that using such a small pixel size makes no sense. In fact, by binning the data 1.5x, the improved signal-to-noise may improve alignment of particles during refinement.

Response: Thanks for your good suggestion. However, due to the limited cryo-EM resources, we are currently unable to reacquire the data. We greatly appreciate your suggestion and will explore it in future studies.

Recommendations

Despite the major issues with the cryo-EM map and model, the overall interpretation of the structural data appears to be correct. The resolution of the (unmodified map) is good enough to fit the AlphaFold model and make an educated guess about the epitope-paratope residues. The way I see it, the authors have two options on how to proceed.

Option 1: Be more forthcoming with the limitations of the current data set.

- The authors should carry out orientation diagnostics of their data and include this as a supplementary figure. This analysis can be performed in the latest version of CryoSPARC. <https://guide.cryosparc.com/processing-data/all-job-types-in-cryosparc/utilities/job-orientation-diagnostics>

Response: Thanks for your valuable suggestion. We performed the orientation diagnostics using the

CryoSPARC v4.2.1, due to the error issues with our rented server for the latest CryoSPARC v4.4.

The outcomes have been included in the new Supplementary Figure 4d and f, shown as follows.

• The authors should show example density for the modified and unmodified regions of the map with the fitted pdb.

Response: We have prepared the following figure to show the modified and unmodified map with the corresponding fitted pdb. The modified map is sharpened compared to the unmodified map.

- Remove domain-level RMSD values. Given the limited resolution, these values are not reliable. Description of movements between domains would be ok to describe at this resolution.

Response: We have revised the description about the domain-level RMSD in the manuscript.

- I would strongly advise using unmodified maps for refinement. There are several nice tools for flexibly fitting models (such as your AlphaFold coordinates) into lower resolution maps (e.g. Namdinator <https://namdinator.au.dk/>).

Response: Done.

Option 2: Obtain a higher-quality experimental structure.

- I appreciate this is probably not feasible given the time and effort required to generate a structure, but there are several ways to alleviate the strong preferred orientation seen in your data. Collecting a tilted data set would be the most straightforward. I can think of one example of a similar sized

complex on graphene oxide that was determined from tilted data:

<https://www.nature.com/articles/s41467-023-38509-2#Sec18>

Response: Thanks for your understanding and good suggestion. However, due to the limited cryo-EM resources, we are currently unable to reacquire the data. We appreciate your suggestion and will explore it in future studies.

Reviewer #2 (Remarks to the Author):

This manuscript describes the isolation, expression, and characterization of two anti-B5 human monoclonal antibodies from B-cells from a previously vaccinated person. One (hMB668) competes with binding to mpox-B5 of a previously published monoclonal (MAb 8AH8AL, ref 26). A cryo-EM structure of hMB668 bound to the first two SCR's of mpox-B5 is also presented and data from that may explain the slightly lower binding of this MAb to the VARV- and CPXV-B5 proteins compared to the MPXV- and VACV-B5 proteins. Authors provide evidence that the other human MAb isolated (hMB621) bound to SCR3-SCR4, but Cryo-EM structure of this complex could not be obtained.

Both MAb showed EV neutralizing activity in the presence of complement and when given 4 hr before and 4 hr after challenge, showed protection of mice in an intranasal mouse challenge model with VACV.

B5 is known to be an important target of a protective immune response and this paper supports this and provides additional important characterization of B5 and human MAbs that may some day be used as part of a cocktail of monoclonal antibodies that could be used for the treatment or prevention of orthopoxvirus infections.

Major points for authors to address:

1. What subtype IgG are these MAbs?

Response: In the process of the MAbs isolation, we used the anti-IgG antibody to sort the IgG⁺ memory B cells. Thus, we can not tell the subtype of the original IgG. However, we constructed the isolated MAbs as IgG1 subtype to evaluate their activities in vitro and in vivo.

2. Do these antibodies inhibit comet plaque formation?

Response: Thanks for your comment. In this study, we utilized the Western Reserve (WR) strain of VACV to evaluate the antiviral effectiveness of hMB621 and hMB668. This strain can form round plaques but not comet-shaped ones. Several published studies have shown that the IHD-J strain of VACV can form comet-shaped plaques. However, due to the inaccessibility of the strain, we were unable to evaluate the inhibitor efficacy of the MAbs. Nevertheless, the MAb 8AH8AL showed a reduction in comet plaque formation¹. Therefore, we speculate that the hMB621 and hMB668, especially hMB668, could inhibit comet plaque formation similar to 8AH8AL, as they recognize a similar epitope. We will explore it in future studies.

3. Need to include the genebank names of the genes expressed. Is the sequence of those expressed proteins the ones shown in Supplementary figure 1A?

Response: The sequences of the expressed MPXV B6 protein and its orthologs are the ones shown in Supplementary Figure 1A. We have included the isolated strains and the accession numbers of these proteins in the Materials and Methods section and the legend of Supplementary Figure 1. The MPXV B6 in the original manuscript is from MPXV_USA_2022_MA001 strain, belonging to clade II. We also added MPXV B6 sequence from the representative strain (Zaire-96-I-16) of clade I and revised Supplementary Figure 1a and b. The new Supplementary Figure 1 is shown below.

Minor points

1. Abstract, line 33. I do not think the data shows the molecular basis of the neutralization and protection of the MAb targeting the B6. At best the cryo EM shows the site of binding /interaction with B6.

Response: We have revised the description.

2. Line 38/39. Would include the word complement since that is a key feature of the neutralization activity.

Response: We have included the word complement.

3. Lines 56. Statements not entirely correct. In U.S., the Jynneos (MVA) vaccine is approved as a vaccine to prevent mpox.

Response: We agree with your opinion. Initially, we intended to say that there are currently no approved MPXV-based vaccines against MPXV infections. To prevent misunderstanding, we have removed the sentence.

4. Lines 64 & 67. While there had been limited clinical data on its effectiveness of MVA to protect against mpox, since the outbreak there has been a lot more data. Authors should also cite more studies that show better effectiveness than the one study they cite.

Response: Thanks for your suggestion. During the review of our manuscript, several studies reported the real-world effectiveness of the MVA vaccine. We have restructured the corresponding description about the MAV vaccine and cited additional references.

5. Line 78. Instead of clinic, would say in humans.

Response: Revised.

6. Line 110. Would not call B5 a poorly characterized protein. This has been an extensively studied orthopoxvirus protein

Response: We have revised the word poorly to incompletely.

7. Line 115. Would end sentence with “in the presence of complement.”

Response: Revised.

8. Line 116. Sentence should read, Moreover, in an intranasal VACV mouse challenge model, the two MAbs exhibited effective protection when given via intraperitoneal (i.p.) injection.

Response: Revised.

9. Line 170. Title should read: Neutralization of antibodies in vitro against VACV infection is complement dependent

Response: Revised.

10. Line 181. Would include what a 10 mg/kg dose represents in mice to better compare to published data in ref 25 which used humanized MAb at 90 ug of purified Mab.

Response: The dose of 10 mg/kg represents 200 µg, due to the mice used weighted about 20 g.

11. Lines 190/191. Given the nature of the challenge model (likely non-lethal challenge dose and other factors), better today, data shows that these MAbs can effectively treat VACV infections in this mouse model.

Response: We have added the phrase “in this mouse model” at the end of the sentence.

12. Line 193. More correct title: Molecular basis of antibodies binding to MPXV B6

Response: Revised.

13. Line 282. Better to say, “Since we could not work with live MPXV ..”

Response: Revised.

14. Line 287. Since neutralization is complement dependent, not sure these can be called “neutralizing epitopes”.

Response: Although the neutralization of MAbs was markedly enhanced in the presence of complement, a certain level of neutralization was still observed without complement (Figure 3). Therefore, these MAbs are proposed as neutralizing antibodies, and their epitopes are considered neutralizing epitopes. Additionally, we also referenced other published studies ^{2,3}, which called related antibodies as neutralizing antibodies.

15. Line 293. Similar to the prior comment, not sure one can say neutralize.

Response: Since the MAbs exhibit inhibitor effect on VACV infection without the addition of complement, we suggest to use neutralize to depict the MAbs.

16. Line 303. Instead of determined, might say obtained.

Response: Revised.

17. Line 316. Occurring instead of increasing

Response: Revised.

18. Figure 2B. legend should explain how competitive binding was done.

Response: Done. For clarity, we have also revised the Figure 2b, shown as below.

19. Figure 3 legend. Would indicate species of complement (rabbit)

Response: Done.

20. Figure 4. How many times was the challenge experiment performed. Seems like it was just once with 5-mice per group?

Response: The experiment was performed once with 5 mice per group, and we have indicated it in the legend.

21. Supplementary figure 1. Need to include gene bank name of proteins being used. Also given interest in Clade 1 and Clade 2 MPXV, would include a representative B5 sequence from each.

Response: Similar to your major point 3, we have included the accession numbers of proteins used and added the representative MPXV B6 sequence from Clade 1.

22. Supplementary Figure 5. None of the colors used in the figure are explained. For example, panel H and I. What is the SCR and what is the antibody?

Response: Following your suggestions and those of Reviewer #1, we have revised Supplementary

Figure 5 as shown below. The colors have been indicated in the legend.

Reference

1 Chen, Z. *et al.* Chimpanzee/human mAbs to vaccinia virus B5 protein neutralize

vaccinia and smallpox viruses and protect mice against vaccinia virus. *Proc Natl Acad Sci U S A* **103**, 1882–1887 (2006).

2 Benhnia, M. R. *et al.* Vaccinia virus extracellular enveloped virion neutralization in vitro and protection in vivo depend on complement. *J. Virol.* **83**, 1201–1215 (2009).

3 Gilchuk, I. *et al.* Cross-neutralizing and protective human antibody specificities to poxvirus infections. *Cell* **167**, 684–694 e689 (2016).

REVIEWERS' COMMENTS

Reviewer #1 (Remarks to the Author):

The authors have satisfactorily addressed my critiques in their revisions.

Reviewer #2 (Remarks to the Author):

This is a revised manuscript where authors have adequately responded my comments. Since my expertise is not in structural biology, Reviewer 1 can comment on the response to their very important review.

The following comments are minor and are provided to further enhance the manuscript.

I now see that authors had included in the methods section information that the constructed antibodies used for their studies were IgG1 subtype. I think it would be useful to better highlight the subtype used to evaluate the activities in vitro and in vivo. Perhaps including the subtype expressed in line 113/114:

In vitro neutralizing assays indicated that the two MAbs (both generated as IgG1 subtypes) showed potent activities against VACV infections in the presence of complement

Also given interest in differences between Clade I and Clade II MPXV, I think it would be useful highlight information now included in the supplementary data. Would suggest including a sentence stating that the expressed B6 protein used in this study (from clade 2 MPXV_USA_2022_MA001) has identical sequence to Clade I strain Zaire-96-I-16.

Line 66/26. Since vaccination coverage of people at highest risk of mpox is so poor, I do not think the following sentence is relevant, "However, MVA has not yet received approval for use in the general population." The paragraph ends with the main point the authors want to make about the need for additional countermeasures.

REVIEWERS' COMMENTS

Reviewer #1 (Remarks to the Author):

The authors have satisfactorily addressed my critiques in their revisions.

Response: Thanks again for your constructive suggestions.

Reviewer #2 (Remarks to the Author):

This is a revised manuscript where authors have adequately responded my comments. Since my expertise is not in structural biology, Reviewer 1 can comment on the response to their very important review. The following comments are minor and are provided to further enhance the manuscript.

1) I now see that authors had included in the methods section information that the constructed antibodies used for their studies were IgG1 subtype. I think it would be useful to better highlight the subtype used to evaluate the activities in vitro and in vivo. Perhaps including the subtype expressed in line 113/114:

In vitro neutralizing assays indicated that the two MAbs (both generated as IgG1 subtypes) showed potent activities against VACV infections in the presence of complement

Response: Thanks for your suggestion. We have revised the sentence.

2) Also given interest in differences between Clade I and Clade II MPXV, I think it would be useful highlight information now included in the supplementary data. Would suggest including a sentence stating that the expressed B6 protein used in this study (from clade 2 MPXV_USA_2022_MA001) has identical sequence to Clade I strain Zaire-96-I-16.

Response: Thanks for your good suggestion. We have added the sentence "Particularly, the expressed B6 protein used in this study (from clade II strain MPXV_USA_2022_MA001) has identical sequence to Clade I strain Zaire-96-I-16 (Supplementary Fig. 1a, b)." in lines 166-168.

3) Line 66/26. Since vaccination coverage of people at highest risk of mpox is so poor, I do not think the following sentence is relevant, "However, MVA has not yet received approval for use in the general population." The paragraph ends with the main point the authors want to make about the need for additional countermeasures.

Response: Thanks for your suggestion. We have deleted the sentence.